

# Projected intensification of sub-daily and daily rainfall extremes in convection-permitting climate model simulations over North America: Implications for future Intensity-Duration-Frequency curves

Alex J. Cannon[1] and Silvia Innocenti[2]

[1]Climate Research Division, Environment and Climate Change Canada, Victoria, Canada
[2]Institut national de la recherche scientifique, Centre Eau Terre Environment, Québec, Canada

**Correspondence:** Alex J. Cannon (alex.cannon@canada.ca)

**Abstract.** Convection-permitting climate models have been recommended for use in projecting future changes in local-scale, short-duration rainfall extremes that are of greatest relevance to engineering and infrastructure design, e.g., as commonly summarized in Intensity-Duration-Frequency (IDF) curves. Based on thermodynamic arguments, it is expected that rainfall extremes will become more intense in the future. Recent evidence also suggests that shorter-duration extremes may intensify

more than longer durations and that changes may depend on event rarity. Based on these general trends, will IDF curves shift upward and steepen under global warming? Will long return period extremes experience greater intensification than more common events? Projected changes in IDF curve characteristics are assessed based on sub-daily and daily outputs from historical and late 21st century pseudo-global warming convection-permitting climate model simulations over North America. To make more efficient use of the short model integrations, a parsimonious Generalized Extreme Value Simple Scaling (GEVSS)

model is used to estimate historical and future IDF curves (1-hr to 24-hr durations). Simulated historical sub-daily rainfall extremes are first evaluated against in situ observations and compared with two high-resolution observationally-constrained gridded products. The climate model performs well, matching or exceeding performance of the gridded datasets. Next, inferences about future changes in GEVSS parameters are made using a Bayesian False Discovery Rate approach. Large portions of the domain experience significant increases in GEVSS location ($> 99\%$ of grid points), scale ($> 88\%$), and scaling exponent

($> 39\%$) parameters, whereas almost no significant decreases are projected to occur ($< 1\%$, $< 5\%$, and $< 5\%$ respectively). The result is that IDF curves tend to shift upward, and, with the exception of the eastern United States, steepen, which leads to the largest increases in return levels for short duration extremes. The projected increase in the GEVSS scaling exponent calls into question stationarity assumptions that form the basis for existing IDF curve projections that rely exclusively on simulations at the daily time scale. When changes in return levels are scaled according to local temperature change, median scaling

rates, e.g., for the 10-yr return level, are consistent with the Clausius-Clapeyron (CC) relation at 1-hr to 6-hr durations, with sub-CC scaling at longer durations and modest super-CC scaling at sub-hourly durations. Further, spatially coherent but small increases in dispersion of the GEVSS distribution are found over more than half of the domain, providing some evidence for return period dependence of future changes in extreme rainfall.



## 1 Introduction

The design of some civil infrastructure – culverts, storm drains, sewers, bridges, etc. – is based on information about local flood extremes with specified low annual probabilities of occurrence (or, equivalently, long return periods). When gauged streamflow data are not available, information about rainfall extremes can instead be used by engineers to infer flood magnitudes for the

return periods of interest. The necessary information on the frequency of occurrence, duration, and intensity of rainstorms is compactly summarized in rainfall Intensity-Duration-Frequency (IDF) curves, and hence IDF curves are a key source of information for water resource and engineering design applications (Canadian Standards Association, 2012). Typical IDF curves summarize the relationship between the intensity and occurrence frequency of extreme rainfall over averaging durations ranging from minutes to a day, usually at local gauging sites. Sub-daily and daily rainfall extremes found in IDF curves are

also featured in building codes (e.g., 15-min and 1-day extreme rainfall is used to estimate roof drainage and loading; National Research Council, 2015).

For the purpose of water resource management and engineering design, it has been stated that "stationarity is dead" (Milly et al., 2008). Increases in atmospheric moisture are expected with anthropogenic global warming as saturation vapour pressure – loosely, the moisture holding capacity of the atmosphere – scales approximately exponentially with temperature following the

Clausius-Clapeyron (CC) relation (~7% per °C). In the absence of other influences, e.g., changes in large-scale circulation, soil moisture availability, etc., the intensity of extreme rainfall should therefore also increase as the atmosphere warms (Trenberth, 2011). While historical increases in local extreme rainfall are difficult to detect due to the relatively small forced signal relative to natural variability, as well as uncertainties due to measurement errors and series length, evidence from observations over large regions and from climate model simulations is largely consistent with widespread thermodynamically-driven intensification of

rainfall extremes (Min et al., 2011; Westra et al., 2013a; Zhang et al., 2013; Pfahl et al., 2017). Moving into the future, even with aggressive mitigation strategies, warming will likely continue over typical design lifetimes due to continued emissions of short- and long-lived greenhouse gases and climate forcing agents (Millar et al., 2017). Attendant changes in rainfall extremes are thus also expected to persist with continued warming.

The reality of climate change, in combination with the long service life of infrastructure, has prompted the incorporation of

future climate projections into the engineering design process. Despite the general expectation that short-duration rainfall extremes will become more intense in the future, there is still substantial uncertainty about the sensitivity of local rainfall extremes to warming (Westra et al., 2014; Pendergrass, 2018). For example, evidence from some idealized convection-permitting model experiments suggests that sub-daily extremes may intensify more than longer durations (O'Gorman, 2015), but the conditions under which this occurs (e.g., sensitivity to microphysics parameterization) are not fully understood. Furthermore, results from

CMIP5 GCMs indicate that more rare extreme daily precipitation events intensify more than less rare events (Kharin et al., 2018), with some indication of return period dependence in sub-daily convection-permitting simulations over small midlatitude domains as well (e.g., Evans and Argueso, 2015; Kuo et al., 2015; Tabari et al., 2016). What will this mean for future IDF curves in North America? Will they shift upward and steepen and will changes in risk depend on return period?



Complicating the transfer of information from climate model simulations to future IDF curves is the historical reliance on (1) climate models with parameterized convection and (2) extrapolation of information on simulated daily extremes to sub-daily extremes. In the first case, short-duration, local-scale rainfall extremes are mostly generated by convective storm systems that are not resolved by most climate models (e.g., those with parameterized convection). Credible projections of localized sub-daily

extreme rainfall may require high-resolution convection-permitting climate models (Kendon et al., 2017). In the second case, prior unavailability of short-duration precipitation outputs from climate models has meant that observed relationships between long and short durations have been used to extrapolate changes at the daily time scale to sub-daily time scales. For instance, Srivastav et al. (2014) used equidistant quantile mapping to statistically downscale and temporally disaggregate daily global climate model outputs to IDF curves at station locations. Assuming a stationary relationship between durations necessarily

constrains relative changes in shorter duration extremes to largely match those at longer durations.

Direct investigation of sub-daily rainfall extremes, and hence IDF curves, from convection-permitting climate models may therefore provide an avenue forward for the water resource management and engineering community. However, integrations of such computationally expensive models are typically short, at most between one to two decades, which makes robust estimation of rare extremes difficult; the high computational expense has also limited their application to small domains. Zhang et al.

(2017) suggested that the use of convection-permitting models, in combination with advanced statistical methods that make better use of short records, may be required to reliably project future changes in short-duration rainfall extremes. Based on this recommendation, the current study links projected changes in sub-daily rainfall extremes from a convection-permitting climate model with changes in specific characteristics of IDF curves using a parsimonious statistical model. Unlike previous studies, which have focused on relatively small domains and/or very short integrations (e.g., Evans and Argueso, 2015; Kuo et al.,

2015; Tabari et al., 2016), the focus here is on decadal simulations for a continental domain covering most of North America. The main goals of the study are to (1) assess whether there is evidence for a shift upward and steepening of IDF curves under global warming; (2) determine whether changes depend on return period; and finally (3) to link projected changes in IDF curve return levels to the magnitude of local warming.

To this end, hourly 4-km rainfall outputs from historical and end-of-century pseudo-global warming convection-permitting

simulations by the Weather Research and Forecasting (WRF) model (Rasmussen et al., 2017; Liu et al., 2017) are used in conjunction with a parsimonious Generalized Extreme Value Simple Scaling (GEVSS) model (Nguyen et al., 1998; Van de Vyver, 2015; Blanchet et al., 2016; Mélèse et al., 2018) to estimate historical and future IDF curves (1-hr to 24-hr durations) over a domain covering northern Mexico, the conterminous United States, and southern Canada. The GEVSS model leverages information from multiple durations to characterize relationships between the frequency of occurrence and duration of extreme

rainfall intensities. Assuming that the underlying model assumptions are met, "borrowing strength" by pooling data from different durations provides more robust estimates of GEV distribution parameters than standard unpooled estimates (Innocenti et al., 2017). Furthermore, the GEVSS model parameterization provides a straightforward framework to make inferences about future changes in IDF curve characteristics. For example, projected increases in the GEVSS temporal scaling exponent lead to greater intensification of shorter duration extremes relative to longer durations, whereas increases in the dispersion

of the GEVSS distribution lead to greater intensification of long return period extremes relative to shorter return periods.



Statistical model assumptions and simulated historical sub-daily rainfall extremes are evaluated using two high-resolution gridded observational products and in situ station observations. Projected changes in GEVSS parameters, and hence IDF curve characteristics, are obtained under a Bayesian framework, with inferences made using a False Discovery Rate (FDR) approach to multiple comparisons. Finally, return level projections are expressed as relative changes with respect to local warming – so called "temperature scaling" – to assess adherence to the theoretical CC relation.

The remainder of the paper is structured as follows. Observational data and model simulations are described in Section 2. The GEVSS approach to IDF curve estimation is provided in Section 3 and the Bayesian framework for parameter inference is summarized in Section 4. Simulated short-duration rainfall extremes and goodness-of-fit of the GEVSS model are evaluated in Section 5. Projected changes in GEVSS parameters and IDF curves are summarized in Section 6 along with estimates of temperature scaling of extreme rainfall return levels. Finally, Section 7 provides a discussion of results, conclusions, and suggestions for future research.

## 2 Observations and simulations

Climate model simulations and observational data used in this study are summarized in Table 1. To investigate future changes in extreme short-duration rainfall and associated IDF curves over North America, precipitation outputs from convection-permitting climate model simulations performed by the National Center for Atmospheric Research (NCAR) using WRF model version 3.4.1 (Liu et al., 2017; Rasmussen et al., 2017) are used at their 1-hr archived time step. Outputs from two sets of WRF simulations – a historical control run (CTRL) and a pseudo global warming (PGW) simulation for the end of the 21st century – each on the same $1360 \times 1016$ 4-km grid, are provided over a domain (referred to as HRCONUS) spanning northern Mexico, the conterminous United States, and southern Canada (Figure 1). In both cases, spectral nudging of geopotential height, horizontal wind, and temperature (5 model layers above the top of the boundary layer) is applied at spatial scales greater than ~2000 km and with an $e$-folding time of ~6-hr. Boundary conditions for the CTRL simulation are given by the European Centre for Medium-Range Weather Forecasts ERA-Interim reanalysis (Dee et al., 2011) for the period 1 October 2000 to 30 September 2013. The PGW simulation uses the same ERA-Interim boundary conditions, but with all variables perturbed with the climate change signal from an ensemble of Coupled Model Intercomparison Project phase 5 (CMIP5) (Taylor et al., 2012) global climate models taken between 1976-2005 and 2071-2100 under the Representative Concentration Pathway (RCP8.5) scenario (Meinshausen et al., 2011).

Importantly, the experimental design – including PGW and spectral nudging – suppresses the influence of internal variability, which would otherwise make detection of forced changes more difficult. It is thus mainly able to isolate the thermodynamic climate change response over the domain. While vertical temperature structure and baroclinicity can be modified by the PGW perturbation, substantive changes in large-scale circulation are not considered (Liu et al., 2017; Prein et al., 2017c). However, decomposition of the forced response of daily-scale extreme rainfall in CMIP5 models into thermodynamic and dynamic components suggests that the dynamic contribution over Canada and the United States is small (Pfahl et al., 2017). Furthermore, by focusing on the late 21st century and RCP8.5 forcings, the PGW experiment is exposed to a relatively large warming signal



**Table 1.** Summary of observational data and climate model simulations used in the study.

| Dataset | Years | Durations | Spatial | Notes |
|---|---|---|---|---|
| ECCC IDF v2.30; ECCC (2014) | Variable | 5-min, 10-min, 15-min, 30-min, 60-min, 2-hr, 6-hr, 12-hr, 24-hr | 488 stations | Canada-wide TBRG |
| CMORPH CRT V1.0; Xie et al. (2017) | 1998-2015 | 1-hr, 3-hr, 6-hr, 12-hr, 24-hr | 0.073° | Merged satellite/gauge |
| MSWEP V2; Beck et al. (2017a, b) | 1979-2016 | 3-, 6-, 12-, 24-hr | 0.1° | Merged satellite/gauge/reanalysis |
| WRF CTRL; Liu et al. (2017) | 2001-2013 | 1-hr, 3-hr, 6-hr, 12-hr, 24-hr | 4-km | ERA-Interim boundary |
| WRF PGW; Liu et al. (2017); Rasmussen et al. (2017) | 2001-2013* | 1-hr, 3-hr, 6-hr, 12-hr, 24-hr | 4-km | * Perturbed ERA-Interim boundary (CMIP5 RCP8.5 2071-2100 - 1976-2005) |

relative to the CTRL simulation (global mean temperature change of +3.5°C), which will also tend to enhance detectability of local-scale changes. Further details on the simulations are provided by Liu et al. (2017) and Rasmussen et al. (2017).

Precipitation outputs from the WRF CTRL simulations over the United States have been evaluated against observations in several studies (e.g., Liu et al., 2017; Dai et al., 2017; Prein et al., 2017c, a; Raghavendra et al., 2018). To extend these results,
the focus here is instead on the Canadian portion of the domain and the range of sub-daily to daily extremes communicated in IDF curves. Annual precipitation maxima at short durations are driven by rainfall, and hence the evaluation deals exclusively with extremes generated by rainstorms. In Canada, national IDF curves are disseminated by Environment and Climate Change Canada (ECCC, 2014) at more than 500 locations with long tipping bucket rain gauge (TBRG) records; 488 stations fall within the WRF HRCONUS domain (Figure 1) and are used in this study. TBRG record lengths range from 10-yr to 81-yr, with a mean
length of 25-yr. Information on the observing program, quality control, and quality assurance methods is provided in detail by Shephard et al. (2014). IDF curves are derived from annual maximum rainfall extremes for accumulation durations ranging from 5-min to 24-hr; model evaluation in this study will also focus on these data. For the WRF CTRL and PGW simulations, annual maximum rainfall intensities are calculated at land grid points for accumulation durations from 1-hr to 24-hr. The first partial calendar year of the CTRL and PGW simulations is treated as spin-up and is not used in any calculations.

As a complement to the in situ observations, data from two observationally-constrained gridded precipitation products are also considered. CMORPH CRT V1.0 is a near-global, reprocessed and bias corrected satellite precipitation product with 30-min temporal and ~8-km spatial resolution (1998-2015) (Xie et al., 2017). Over land, CMORPH CRT adjusts raw CMORPH satellite precipitation estimates towards gauge analyses using a probability density function matching algorithm. MSWEP V2 is a global merged product that combines information from satellite, reanalysis, and gauge precipitation estimates at a 3-hr
temporal and 0.1° spatial resolution (1979-2016) (Beck et al., 2017a, b). For consistency with the WRF CTRL simulations,



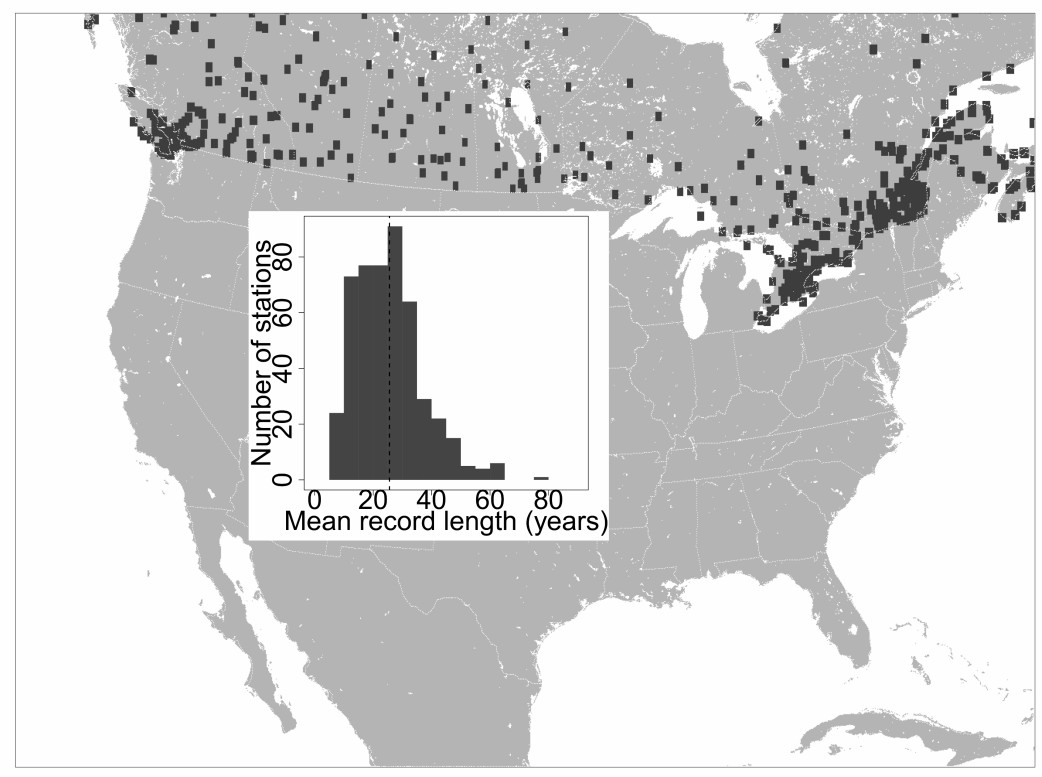

**Figure 1.** Map showing the NCAR HRCONUS WRF land mask and locations (rectangles) of the 488 IDF curve TBRG stations in Canada. The inset plot shows a histogram of mean record length (all IDF curve durations) at the stations; the median (vertical dashed line) is 25 years.

annual maximum rainfall intensities are calculated for accumulation durations ranging from 1-hr to 24-hr for CMORPH and from 3-hr to 24-hr for MSWEP.

## 3   IDF curves and the GEVSS distribution

IDF curves provided by ECCC (2014) summarize the relationship between observed annual maximum rainfall intensity for specified frequencies of occurrence (2-yr, 5-yr, 10-yr, 25-yr, 50-yr, and 100-yr return periods, i.e., 0.5, 0.8, 0.9, 0.96, 0.98, and 0.99-quantiles) and durations (5-min, 10-min, 15-min, 30-min, 60-min, 2-hr, 6-hr, 12-hr, and 24-hr). Because TBRG records rarely exceed 50 years in length (Figure 1), return value estimates at long return periods rely on statistical extrapolation guided by extreme value theory (Coles, 2001). In Canada, official IDF curves are constructed by first fitting the Gumbel distribution, i.e., the extreme value type I form of the GEV distribution, to annual maximum rainfall intensity series at each site for each duration (Hogg et al., 1989). At the majority of stations, the actual curves are then based on best fit linear interpolation equations between log-transformed duration and log-transformed quantiles for each of the specified return periods. To illustrate, IDF



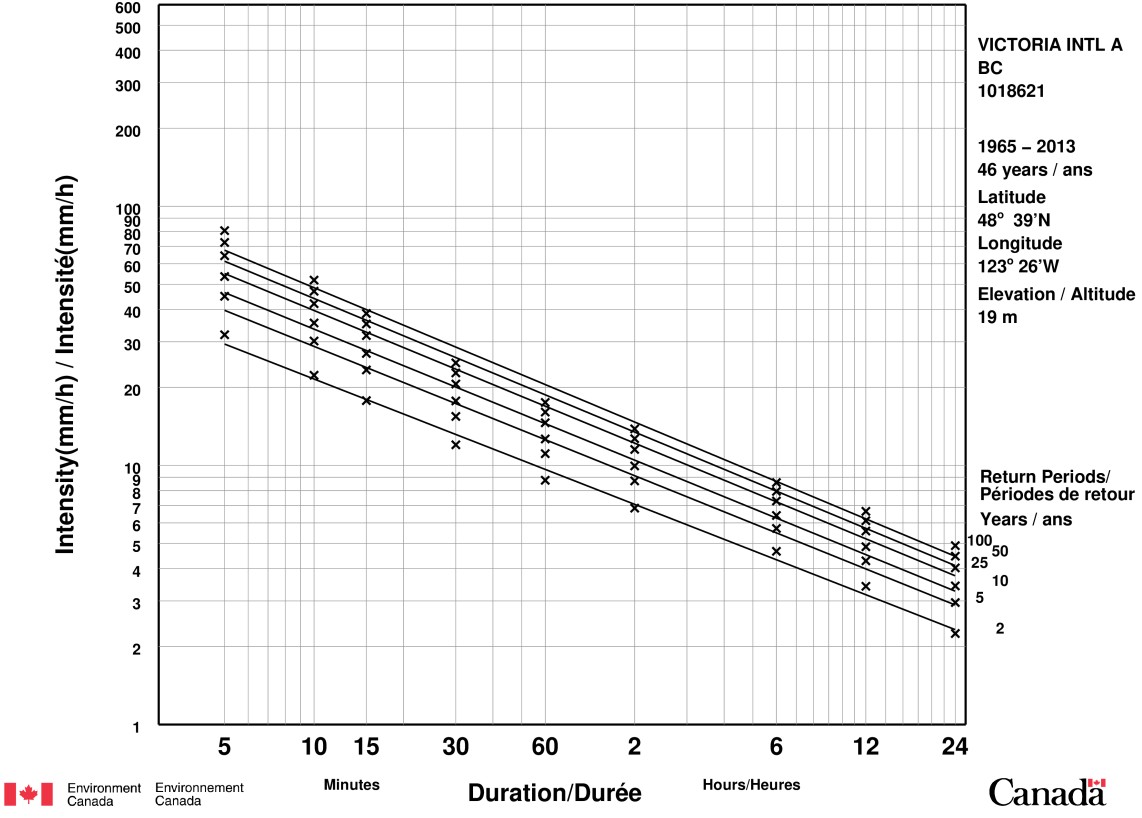

**Figure 2.** Example ECCC IDF data for Victoria Intl A (station 1018621) in British Columbia, Canada. Points (×) show quantiles associated with 2-yr, 5-yr, 10-yr, 25-yr, 50-yr, and 100-yr (from bottom to top) return period intensities estimated by fitting the Gumbel form of the GEV distribution by the method of moments to annual maximum rainfall rate data for 5-min, 10-min, 15-min, 30-min, 60-min, 2-hr, 6-hr, 12-hr, and 24-hr durations (left to right). Lines are from best fit linear interpolation equations between log-transformed duration and log-transformed Gumbel quantiles for each return period.

curves for Victoria Intl A, a station on the southwest coast of British Columbia, Canada, are shown in Figure 2. Points indicate return values of rainfall intensity obtained from the fitted Gumbel distribution for each combination of return period and duration. IDF curves for each return period are based on log-log interpolating equations through these points, and hence plot as straight lines. While IDF curves produced in other national and sub-national jurisdictions may be based on slightly different procedures and assumptions (Svensson and Jones, 2010), results are typically presented in similar fashion and, broadly, share common characteristics.



For simplicity, the study starts with the ECCC IDF curve methodology as its basis. However, modifications to the standard approach are made to accommodate (1) the relatively short 13-yr record lengths provided by the WRF CTRL and PGW simulations, (2) the underlying goal of assessing changes in general IDF curve characteristics (i.e., shifts and/or changes in IDF curve slope), and (3) the current state-of-the-art around the analysis of rainfall extremes. More specifically, the two-step

Gumbel/log-log interpolating equation approach is replaced with single-step estimation of IDF curves using a GEV simple scaling (GEVSS) distribution. Observational evidence suggests that daily rainfall extremes follow an extreme value distribution with a heavy upper tail (Papalexiou and Koutsoyiannis, 2013). Hence, a three-parameter GEV distribution, which includes the heavy-tailed type II extreme value or Fréchet form of the GEV distribution, is used rather than artificially restricting the GEV shape parameter to be zero (i.e., by using the two-parameter extreme value type I or Gumbel distribution). Separate GEV

distribution parameters for each duration of interest, combined with separate interpolating equations for each quantile, leads to a very large number of statistical parameters that need to be estimated from relatively short WRF simulations. Despite the large forced signal and PGW experimental design, the limited sample size necessitates making efficient use of the available data. Hence, to reduce the overall number of distributional and regression parameters that need to be estimated, an aggregated GEVSS distribution is instead fitted to pooled annual maxima for all durations. The GEVSS model is based on the application

of the simple scaling hypothesis – an empirical power-law relation that links the distributions of rainfall intensities at different durations – to the GEV distribution.

Use of the GEV distribution is motivated by the extreme value theorem, which states that the GEV is the only possible limit distribution for the maxima of a sequence of independent and identically distributed random variables (Coles, 2001). Defining $I_{D_0}$ as the random variable of annual maximum rainfall intensity (mm/hr) for an arbitrary reference duration, in this case

$D_0 =$24-hr, it is assumed that samples of annual maxima are distributed according to the GEV distribution

$$Pr(I_{D_0} \leq x) = \begin{cases} \exp\left\{-\left(1+\xi_0 \frac{x-\mu_0}{\sigma_0}\right)^{-1/\xi_0}\right\} & \text{if } \xi_0 \neq 0 \\ \exp\left\{-\exp\left(-\frac{x-\mu_0}{\sigma_0}\right)\right\} & \text{if } \xi_0 = 0 \end{cases} \tag{1}$$

where $\mu_0$, $\sigma_0 > 0$, and $\xi_0$ are, respectively, the GEV location, scale, and shape parameters; $\xi_0 = 0$ corresponds to the type I extreme value distribution (Gumbel form), $\xi_0 > 0$ to the type II (Fréchet form), and $\xi_0 < 0$ to the type III (Weibull form). Quantiles associated with the $T$-yr return period $T = 1/\left[1 - Pr(I_{D_0} \leq x)\right]$ are determined by inverting the GEV cumulative

distribution function given by equation 1.

To incorporate other durations, simple scaling makes the assumption that

$$I_D \stackrel{\text{dist}}{=} \left(\frac{D}{D_0}\right)^{-H} I_{D_0} \tag{2}$$



**Figure 3.** (a) Observed GEVSS IDF curves at Victoria Intl A (1018621) (cf. Figure 2). Hypothetical IDF curves resulting from (b) increases of $\Delta\mu_0 = +23\%$ and $\Delta\sigma_0 = +23\%$ (i.e., no change in dispersion $\sigma_0/\mu_0$); (c) increases of $\Delta\mu_0 = +23\%$ and $\Delta\sigma_0 = +30\%$ (i.e., an increase in dispersion $\sigma_0/\mu_0$); (d) increases of $\Delta\mu_0 = +23\%$, $\Delta\sigma_0 = +23\%$, and $\Delta H = +0.03$; and (d) increases of $\Delta\mu_0 = +23\%$, $\Delta\sigma_0 = +30\%$, and $\Delta H = +0.03$. The matrix plots that accompany (b) - (d) show the associated % changes in return levels for each duration: (b) with constant $H$, increasing $\mu_0$ and $\sigma_0$ without changing the dispersion leads to relative increases in return levels for all durations that match the relative changes in the underlying parameters; (c) increasing dispersion leads to return period-dependence of changes, with larger relative increases evident at longer return periods; (d) increasing $H$ steepens the IDF curves, which leads to duration-dependence of changes; and (e) increases in both $H$ and dispersion result in greater intensification at longer return periods and shorter durations. Note: values in (e) are based on domain mean values from Section 6.





where $0 < H < 1$ is a scaling exponent and $\overset{\text{dist}}{=}$ means equality of distributions (Gupta and Waymire, 1990); for the GEV distribution (Nguyen et al., 1998), simple scaling implies that

$$\mu_D = \left(\frac{D}{D_0}\right)^{-H} \mu_0; \; \sigma_D = \left(\frac{D}{D_0}\right)^{-H} \sigma_0; \; \xi_D = \xi_0 = \xi. \tag{3}$$

The resulting GEVSS distribution for annual maximum rainfall intensities at different durations can be described by four parameters: location $\mu_0$ and scale $\sigma_0$ parameters associated with the reference duration, a temporal scaling exponent $H$ used to scale the reference location and scale parameters to other durations, and a shared shape parameter $\xi$ for all durations. This leads to a simple expression for IDF curves at any duration and return period

$$i_{D,T} = \left(\frac{D}{D_0}\right)^{-H} \left\{ \mu_0 - \frac{\sigma_0}{\xi} \left[ 1 - \left( -\log\left[1 - \frac{1}{T}\right] \right)^{-\xi} \right] \right\} \tag{4}$$

where $i_{D,T}$ is the return level estimate for duration $D$ and the $T$-yr return period (Mélèse et al., 2018). In addition, changes in each parameter can be linked to changes in specific characteristics of IDF curves. For reference, hypothetical examples, based on the data shown in Figure 2, are provided in Figure 3. The scaling exponent $H$ controls the common slope, in log-log space, of linear IDF curves for each quantile. Larger values lead to steeper IDF curves. Hence, increases in $H$ provide direct evidence for stronger intensification of shorter duration rainfall extremes than longer durations. The location $\mu_0$ and scale $\sigma_0$ parameters control the vertical positioning (and hence changes lead to shifts) of the IDF curves. Furthermore, with constant $\xi$, changes in the non-dimensional dispersion coefficient $\sigma_0/\mu_0$ of the GEV distribution result in relative changes in return level that depend on return period, i.e., for a given duration, an increase in dispersion means that relative changes of more rare events will increase more than less rare events.

## 4 Parameter inference

Inferences about GEVSS parameters are made using a Bayesian framework (Van de Vyver, 2015; Mélèse et al., 2018). Posterior parameter distributions are obtained at each grid point from pooled 1-hr, 3-hr, 6-hr, 12-hr, and 24-hr annual maximum rainfall intensities using the Metropolis-Hastings Markov Chain Monte Carlo (MCMC) algorithm (Kruschke, 2015). The Bayesian GEVSS model for CTRL and PGW rainfall intensities is defined as

$$\begin{aligned} I_{D_{\text{CTRL}}} &\sim GEVSS\left(\mu_{0_{\text{CTRL}}}, \sigma_{0_{\text{CTRL}}}, \xi, H_{\text{CTRL}}\right) \\ I_{D_{\text{PGW}}} &\sim GEVSS\left(\mu_{0_{\text{PGW}}}, \sigma_{0_{\text{PGW}}}, \xi, H_{\text{PGW}}\right) \\ D_0 &= 24\text{-hr}, \; D = \{1\text{-hr}, 3\text{-hr}, 6\text{-hr}, 12\text{-hr}, 24\text{-hr}\} \end{aligned} \tag{5}$$





$$\mu_{0_{\mathrm{CTRL,PGW}}} \sim U\left(0.01\,\bar{i}_{0_{\mathrm{CTRL}}}, 4\,\bar{i}_{0_{\mathrm{CTRL}}}\right)$$
$$\sigma_{0_{\mathrm{CTRL,PGW}}} \sim U\left(0.01\,\bar{i}_{0_{\mathrm{CTRL}}}, 4\,\bar{i}_{0_{\mathrm{CTRL}}}\right)$$
$$\xi \sim N\left(0.114, 0.045\right)$$
$$H_{\mathrm{CTRL,PGW}} \sim U\left(0, 1\right)$$

(6)

where $\bar{i}_{0_{\mathrm{CTRL}}}$ is the sample mean 24-hr annual maximum rainfall intensity for the WRF CTRL simulation. A relatively broad uniform prior distribution, with limits constrained to be positive multiples of $\bar{i}_{0_{\mathrm{CTRL}}}$, is used for both the location $\mu_0$ and scale $\sigma_0$ parameters of the CTRL and PGW simulations. This choice is informed by past work showing (1) that end-of-century

projected changes in annual rainfall extremes by CMIP5 models, which scale similarly to $\mu_0$ and $\sigma_0$, are expected to be less than half the upper limit of the prior (Kharin et al., 2013; Toreti et al., 2013; Cannon et al., 2015) and (2) that observational estimates of the dispersion $\sigma_0/\mu_0$ for daily rainfall extremes in North America is on the order of ~0.2 (Koutsoyiannis, 2004). While this implies that a narrower prior could be used for $\sigma_0$, a more conservative choice is adopted here. Following, Kharin et al. (2013) and related work, it is assumed that the shape parameter $\xi$ is the same in the CTRL and PGW periods (see below

for more details). The normal prior distribution used for $\xi$ follows from an analysis of daily station observations by Papalexiou and Koutsoyiannis (2013), who found that $\xi$ varies globally in a narrow range $0 < \xi < 0.23$. Finally, the uniform distribution between 0 and 1 for $H$ follows from Van de Vyver (2015).

   Posterior distributions for GEVSS parameters at each grid point are estimated from 100,000 MCMC samples taken following a burn-in period of 10,000 samples. Standard diagnostics (e.g., Geweke, Ratery-Lewis, and Heidelberg-Welch tests; Plummer

et al., 2006) are used to assess convergence of the chain; these are complemented by spot visual inspections at randomly selected grid points. Because of the high model resolution, large domain, and associated storage cost, the MCMC chain is thinned to 1,000 samples by saving every 100th sample. All subsequent results are based on the thinned chain. The independence likelihood is used for the GEVSS model. Hence, the model is, from a strict standpoint, misspecified as simulated annual rainfall extremes for different durations can be generated by the same storm system. Implications of this lack of independence

on the posterior distributions are examined via Monte Carlo simulation in the supplemental material (Figure S1). Results suggest that posterior distributions for $\mu_0$ and $\sigma_0$ are slightly too narrow when the independence assumption is violated, but those for $\xi$ and $H$ are reliable. To test the stationarity assumption for $\xi$, separate models in which $\xi$ was free to differ between the CTRL and PGW simulations are also considered. Modified deviance information criterion $\mathrm{DIC}^*$ differences between the nonstationary and stationary models (Spiegelhalter et al., 2014) and posterior distributions of the $\xi_{\mathrm{PGW}} - \xi_{\mathrm{CTRL}}$ differences

for the nonstationary model (Figure S2) confirm that the assumption of constant shape is reasonable. Subsequent results are thus reported for the stationary $\xi$ model; unless noted otherwise, results are based on posterior means.

## 5   Historical model evaluation

Prior to assessing projected changes in extreme rainfall and IDF curve characteristics, evaluations are first conducted to assess whether (1) extreme rainfall in the WRF CTRL simulation is well-simulated; (2) whether the GEVSS statistical model





assumptions are met; and, finally, whether (3) WRF CTRL IDF curve estimates based on the GEVSS statistical model are consistent with those from observations. As noted above, simulated rainfall extremes from the WRF CTRL simulation have already been evaluated against station data over the United States (e.g., Figures S2-S4 in Prein et al., 2017c, which led the authors to conclude that "The control simulation is able to reproduce the observed intensity and frequency of extreme hourly

precipitation in most parts of CONUS"). The focus here is thus on the Canadian portion of the domain.

Annual maxima from in situ observations arguably represent the most accurate and reliable estimates of extreme rainfall, despite instrumental uncertainty and limited spatiotemporal coverage. For this reason, extremes from the WRF CTRL simulation are first assessed against data from the 488 IDF curve TBRG stations shown in Figure 1. For comparison, results are also provided for the CMORPH and MSWEP observationally-constrained gridded datasets (Table 1), bearing in mind that both

products are directly informed by station observations within the WRF domain and are thus not strictly independent from the verification data. Also, it is assumed that the 4-km WRF, 0.073° CMORPH, and 0.1° MSWEP grids are of sufficiently high resolution that they can be meaningfully compared against in situ data, notwithstanding differences in interpolation/gridding methods, grid area (nominally 16 km$^2$ to ~80 km$^2$ at the median station latitude), and inherent issues with grid-to-point comparisons.

Because extreme rainfall is a non-negative quantity with substantial spatial variation over the domain, model performance statistics used for evaluation are based on the accuracy ratio $\mathrm{AR}_i = \hat{y_i}/y_i$ or, equivalently, relative error (or bias) $\mathrm{RE}_i = \mathrm{AR}_i - 1$, where $y_i$ and $\hat{y_i}$ are observed and modelled values for $i = 1 \dots N$ locations. Statistics include the mean logarithmic accuracy ratio, which summarizes relative bias,

$$\mathrm{MLAR} = \frac{1}{N} \sum_{i=1}^{N} \log_{10} \left( \mathrm{AR}_i \right), \tag{7}$$

and mean absolute logarithmic accuracy ratio, which summarizes overall magnitude of relative error,

$$\mathrm{MALAR} = \frac{1}{N} \sum_{i} \left| \log_{10} \left( \mathrm{AR}_i \right) \right|. \tag{8}$$

Notably, LAR-based statistics, unlike those based on untransformed RE, are symmetric measures of relative error, i.e., proportional errors of $1/10$ and $10$ are assigned equal magnitudes (-1 and +1 respectively).

Before summarizing performance in terms of MLAR and MALAR, maps of RE for the empirical 90th percentile (10-yr

return level) of annual maximum rainfall between the gridded datasets and station observations are shown in Figure 4 for 1-hr, 6-hr, and 24-hr durations. A permutation test (null hypothesis of equality) based on 5,000 random samples is used to estimate the statistical significance of the RE values. Due to the short WRF record length, empirical return levels for return periods longer than 10-years are not considered. For all stations and grid points, empirical quantiles are calculated based on the entire period of record, which differs in length at each station and product. While this will result in some level of unknown, station-

dependent bias, there is limited evidence to suggest that historical trends in annual maximum short-duration rainfall intensity





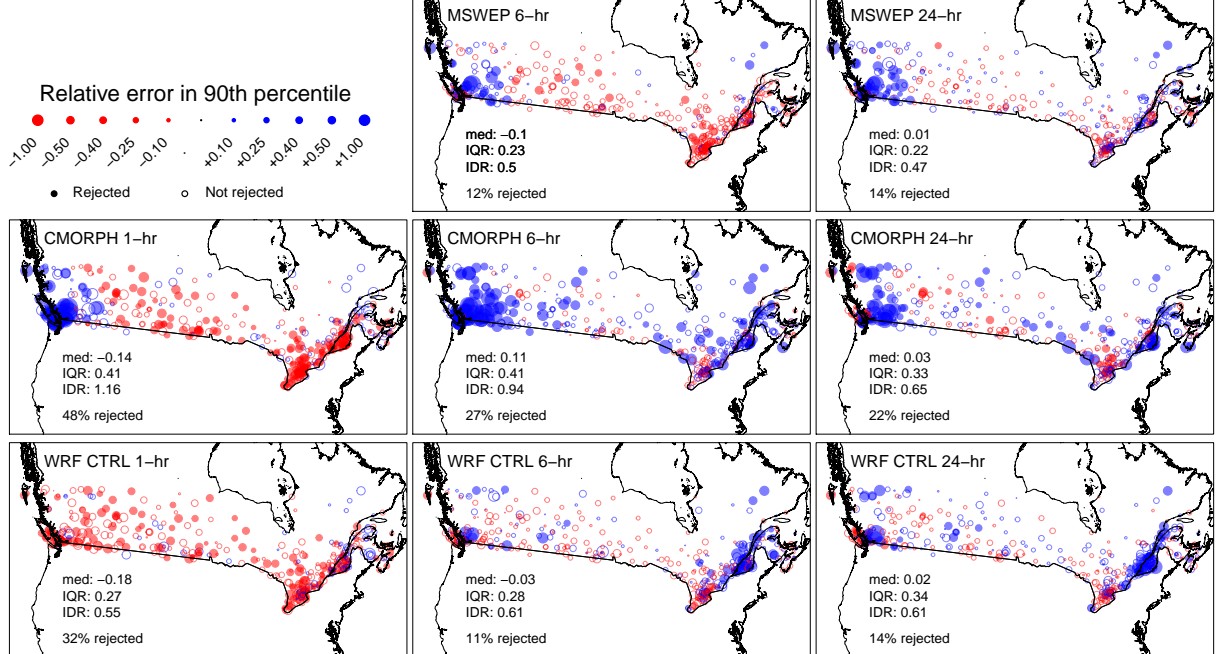

**Figure 4.** Relative error (RE) in the empirical 90th percentile (10-yr return level) of annual maximum rainfall intensities between each of MSWEP (top row), CMORPH (middle row), and WRF CTRL (bottom row) gridded datasets and station observations for 1-hr (left column), 6-hr (middle column), and 24-hr (right column) accumulation durations. Summary statistics reported in each figure include the median (med), interquartile range (IQR), and interdecile range (IDR) of RE, as well as the percentage of locations exhibiting statistically significant values of RE. Statistical testing is performed at the 0.05 significance level.

are detectable at individual stations (Shephard et al., 2014; Barbero et al., 2017); an argument can thus be made in favour of using as much data as possible for estimation, rather than harmonizing the periods of record.

Figure 4 indicates that performance of WRF in terms of RE reaches or, in some cases, exceeds that of the best observationally-constrained product. Spatially, RE for WRF and the two gridded observational products is generally low over the interior of

5 the domain for 6-hr and 24-hr durations. Bias is largest over coastal regions and the western Cordillera, where both the simulations and gridded observations tend to overestimate the 10-yr return level. Notably, MSWEP and WRF perform better than CMORPH at these two durations. For the 24-hr (6-hr) duration, significant RE values are found at 14% (12%) of stations for MSWEP, 14% (11%) for WRF, and 22% (27%) for CMORPH. Performance at the shortest 1-hr duration degrades for both CMORPH and WRF – the two gridded products with a sampling frequency < 3-hr – with significant RE found at 48% and 32%

10 of stations, respectively. In both cases, 10-yr return levels tend to be underestimated, except in the west for CMORPH, with median RE values of -0.14 and -0.18 for CMORPH and WRF, respectively. Note, however, that results are more consistent for WRF (interquartile and interdecile ranges of 0.27 and 0.55 versus 0.41 and 1.16 for CMORPH).

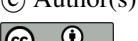



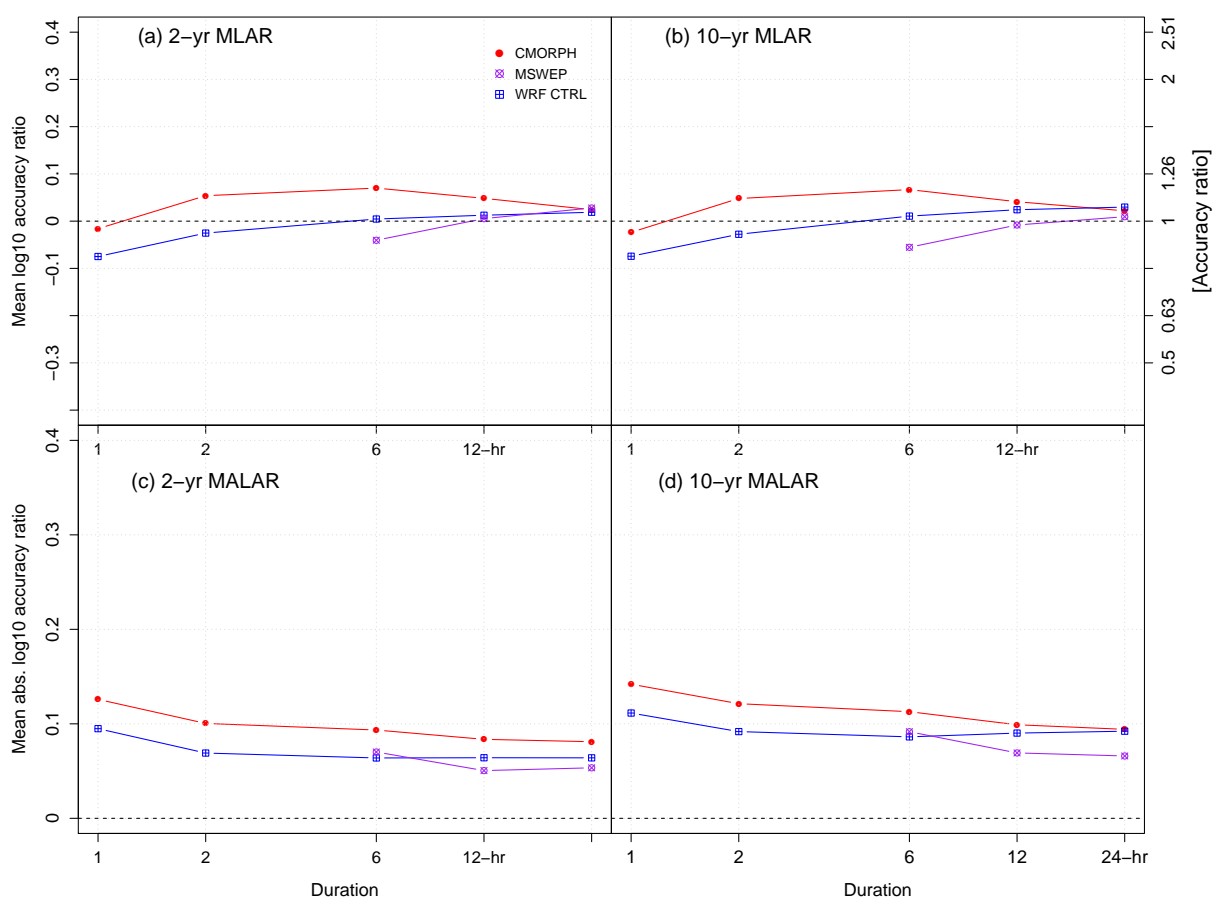

**Figure 5.** Summaries of MLAR and MALAR for the empirical 50th (2-yr return level) and 90th percentile (10-yr return level) of annual maximum rainfall intensities between each of MSWEP, CMORPH, and WRF CTRL gridded datasets and station observations for 1-hr to 24-hr accumulation durations. Perfect performance is indicated by the horizontal dashed lines.





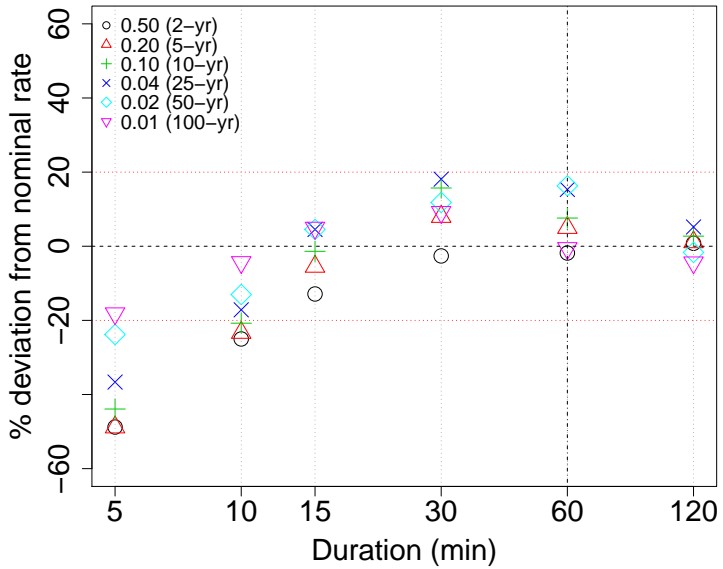

**Figure 6.** Posterior predictive checks for sub-hourly durations extrapolated from GEVSS distributions fitted to observed annual maxima for durations ≥60-min. Values are relative deviations from nominal exceedance rates for 2-yr to 100-yr return level estimates (i.e., 50% to 1% chance of exceedance, respectively). Perfect reliability is indicated by the horizontal black dashed line; deviations of ±20% are indicated by horizontal red dotted lines.

MLAR and MALAR, which provide aggregated measures of performance over the in situ station network, are shown in Figure 5 for quantiles associated with the 2-yr and 10-yr return levels for each of the 1-hr, 2-hr, 6-hr, and 24-hr durations. The pattern of relative bias across durations for WRF and the two sets of gridded observations, expressed in terms of MLAR, is similar for the 2-yr and 10-yr return levels. All datasets show similar values of MLAR at the 24-hr duration, whereas WRF

5   outperforms CMORPH for durations from 2-hr to 12-hr, and MSWEP at the 6-hr duration. WRF underestimates more than CMORPH at the shortest 1-hr duration, but the overall level of bias is still modest for both datasets. However, when contributions from both systematic and unsystematic errors are taken into consideration via MALAR, WRF is found to outperform CMORPH for both return levels and all durations. MSWEP performs best for the 12-hr and 24-hr durations, but is matched by WRF at the 6-hr duration.

10   In addition to verifying that WRF can reliably simulate sub-daily rainfall extremes, the ability of the GEVSS statistical model to describe the distribution of annual rainfall maxima must also be checked. One of the advantages of the GEVSS model is that it can be used to extrapolate from available durations to shorter durations, e.g., from simulated 1-hr data to sub-hourly durations needed in IDF curves or building codes. However, the ability to extrapolate depends strongly on the GEVSS goodness-of-fit, especially for the shortest durations where the simple scaling hypothesis may no longer hold (Innocenti et al., 2017). If the

15   GEVSS model is consistent with observations, then observed exceedance probabilities of the predicted quantiles should match




nominal values. For example, 10% of station observations should exceed the predicted 10-yr return levels (the 90th percentiles), and so forth. Results from posterior predictive checks of the GEVSS distribution for observed annual maxima are shown in Figure 6. In this case, the GEVSS distribution is fitted to pooled 1-hr, 2-hr, 6-hr, 12-hr, and 24-hr duration observations at each station, and predictive quantiles are computed and compared against observations for sub-hourly data not used to fit the model.

As expected, results are consistent with observations for the 1-hr and 2-hr durations used to fit the GEVSS model. The model continues to perform reliably when extrapolating to the 30-min and 15-min durations, but begins to deviate from the nominal exceedance probabilities for the 10-min duration. Expected exceedance probabilities are underestimated by ~20-55% at the 5-min duration, suggesting that predicted GEVSS return levels are overpredicted; results for the shortest durations should thus either not be used or be treated with caution.

Given findings above that (1) daily and sub-daily rainfall extremes from the WRF model are generally well-simulated and (2) that the GEVSS distribution provides an acceptable fit to station observations for durations >10-min, GEVSS IDF curves derived from WRF outputs should resemble those calculated directly from in situ observations. To verify that this is indeed the case, IDF curve return levels estimated from WRF (and, for reference, CMORPH and MSWEP) are compared with those calculated from station observations. (A separate intercomparison is presented for GEVSS parameters in Figures S3-

S5.) The observational reference in this case is the set of return level estimates from IDF curves disseminated by ECCC (i.e., calculated using the two-step Gumbel/log-log regression methodology outlined in Section 3). Furthermore, uncertainty due to differences in estimation methodology is examined by comparing the ECCC IDF curves with those obtained by fitting the GEVSS distribution to station data. This is done for GEVSS-based estimates from the full set of observed durations (5-min to 24-hr) and a restricted set of durations (1-hr to 24-hr) to match the sampling frequency of the WRF outputs. Results are shown

for relative bias (MLAR) in Figure 7 and for relative error magnitude (MALAR) in Figure 8. Fitting the GEVSS distribution to station data results in IDF curves that are largely similar to the official ECCC IDF curves. When fitting to all observed durations, differences are small, even for sub-hourly durations. When the set of fitted durations is restricted to be $\geq$ 1-hr and the GEVSS distribution is used to extrapolate to sub-hourly durations, estimates are consistent with the ECCC IDF curves down to the 15-min duration; overprediction is evident at the shortest 5-min and 10-min durations. Turning now to GEVSS

IDF curves estimated from WRF CTRL outputs, results indicate that WRF is generally equally consistent or more consistent than CMORPH to ECCC IDF curves for all durations and return levels. The same is true when WRF is compared with MSWEP for durations $\leq$ 6-hr.

## 6   Future projections

The main goal of this study is to investigate how IDF curve characteristics are projected to change in convection-permitting

climate model simulations. From a statistical perspective, this is achieved by making inferences about changes in the parameters of the GEVSS distribution. Inference is based on posterior distributions of differences in parameters between the CTRL and




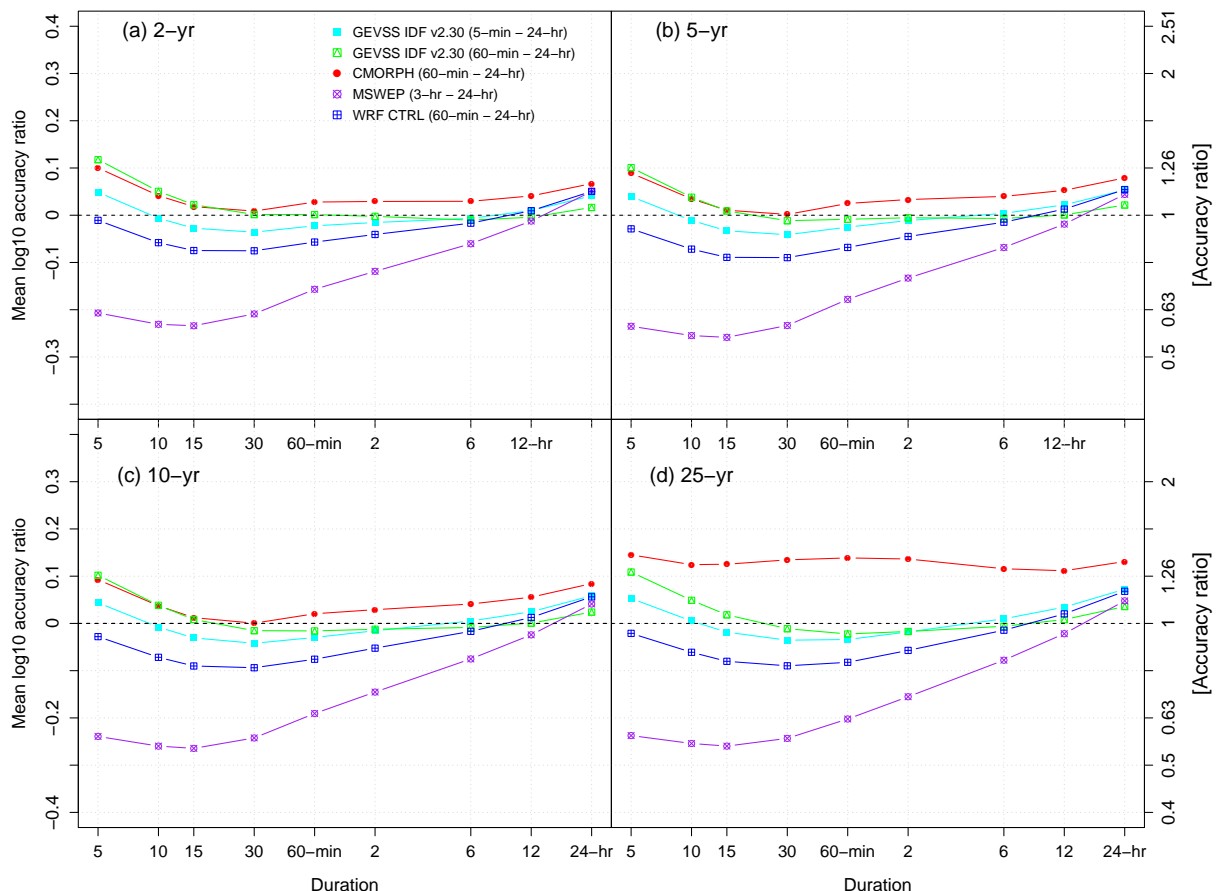

**Figure 7.** Summaries of MLAR for the empirical (a) 50th (2-yr return level), (b) 80th (5-yr return level), (c) 90th (10-yr return level), and 96th (25-yr return level) percentile of annual maximum rainfall intensities between each of MSWEP, CMORPH, and WRF CTRL gridded datasets and ECCC IDF curve v2.30 estimates for 5-min to 24-hr accumulation durations at TBRG stations. For reference, values are shown for GEVSS IDF curve estimates based on 5-min to 24-hr TBRG data and a restricted set of 60-min to 24-hr durations. Perfect performance is indicated by the horizontal dashed lines.





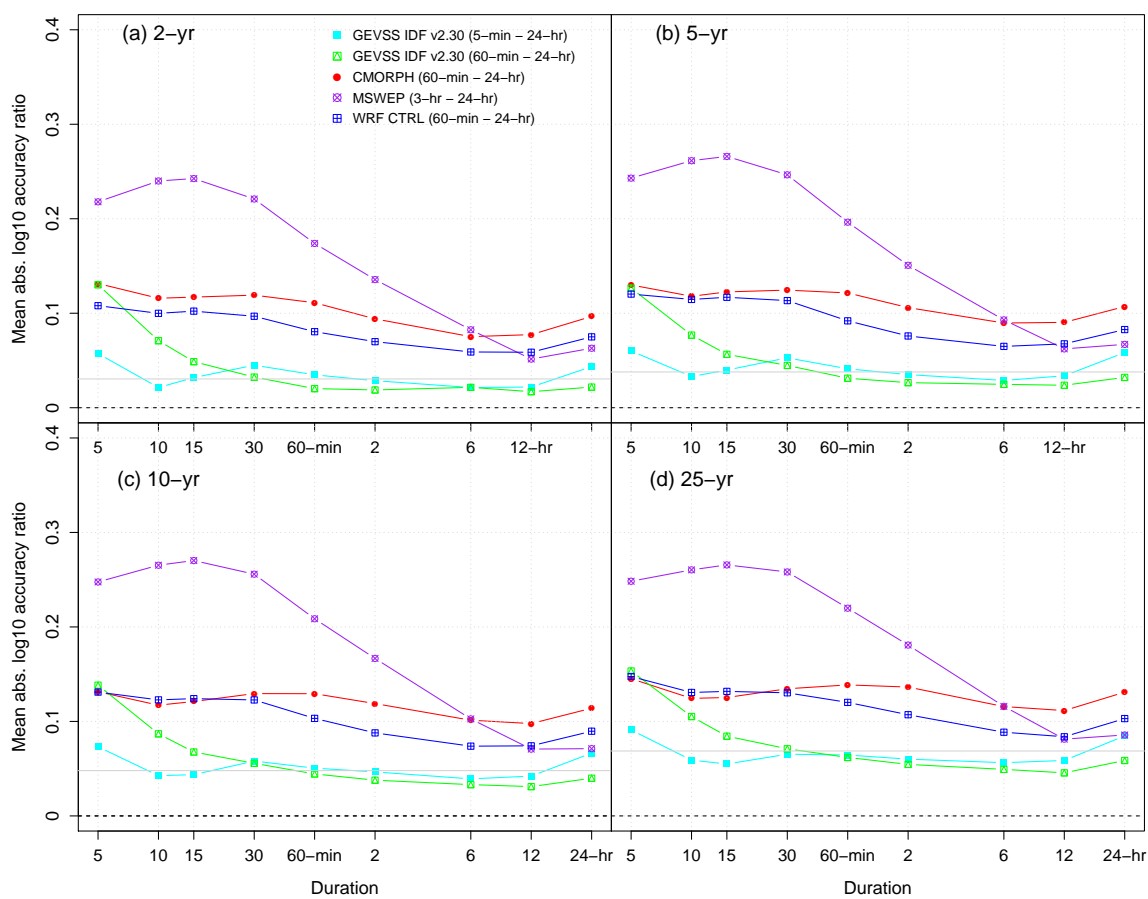

**Figure 8.** As in Figure 7, but for MALAR. For reference, the horizontal gray line indicates the expected MALAR for empirical quantile estimates, based on 25-yr samples (the median record length of TBRG station observations), from a true $GEV(\mu = 1.93, \sigma = 0.64, \xi = 0.10)$ distribution; parameters correspond to median estimates at the 488 IDF curve TBRG stations.





PGW simulation periods. For example, if the goal is to assess whether there is evidence for a steepening of IDF curves in the future

$$\Delta H_{\mathrm{PGW-CTRL}} > 0 \tag{9}$$

then values of the posterior error probability (PEP), defined as $Pr(\Delta H_{\mathrm{PGW-CTRL}} \leq 0)$, are determined from the posterior
parameter distributions at each grid point. To account for the multiple comparisons problem, the PEP values are evaluated
under a Bayesian FDR framework – a Bayesian analogue of the approach recommended by Wilks (2016) – using a global FDR
of 10% following Storey et al. (2003), Käll et al. (2008), and as summarized by Robinson (2017). Based on the PEP, grid points
with sufficient evidence for an increase are identified such that no more than 10% are included by mistake, i.e., the probability
of correctly identifying an increase in a given parameter at a grid point is at least 90%. Conversely, one can evaluate evidence
for decreases in GEVSS parameters by reversing the definition of PEP.

Projected PGW-CTRL changes over the domain are shown in Figure 9. Significant increases in GEVSS location ($> 99\%$ of
grid points), scale ($> 88\%$), and scaling exponent ($> 39\%$) parameters are projected over large portions of the domain, whereas
almost no significant decreases in the GEVSS parameters are projected to occur ($< 1\%$, $< 5\%$, and $< 5\%$ respectively). The
result is that IDF curves tend to shift upward and, with the exception of the eastern US, steepen, which leads to the largest
increases in return values for short duration extremes at the end of the 21st century. For example, at the 1-hr duration, the
median projected increase in the 10-yr return value (Figure 9d) is +38% (+29% lower quartile, +49% upper quartile), versus
+25% (+18%, +33%) for the 10-yr return level of the 24-hr duration. The projected increase in the GEVSS scaling exponent
calls into question stationarity assumptions (i.e., that daily to sub-daily scaling remains the same) that form the basis for existing
IDF curve projections that rely exclusively on simulations at the daily time scale.

These findings are for a strong +3.5°C global warming signal that corresponds to end of century conditions under the RCP8.5
scenario, which limits their general usefulness. Given that there is little evidence to suggest that changes in precipitation
extremes for a given temperature change depend on RCP forcing scenario over North America (Pendergrass et al., 2015),
results are reframed in terms of local scaling with annual mean temperature change. Assuming that the temperature scaling
relationship holds, which may depend on the relative composition of aerosol vs. greenhouse gas forcings (Lin et al., 2016),
future projections of local temperature can then be used to gain information about future return levels of extreme rainfall.
Temperature scaling results for the 1-hr and 24-hr 10-yr return levels are shown in Figures 10a and b, respectively, with
summaries for the other durations shown in Figure 10c. Based on the model evaluation presented in Section 5, results are not
shown for the 5-min and 10-min durations. Median temperature scaling values for 1-hr (7.6%/°C) to 6-hr (6.2%/°C) durations
are consistent with the CC relation (~6%/°C to 8%/°C over the range of temperatures simulated by WRF), although with some
regional variation. Notably, larger scaling magnitudes are found over coastal regions and a region extending northward from
the Baja peninsula into the Great Basin. Scaling rates in the interior of the continent – the Great Plains and Central Lowland
– tend to be smaller. Sub-hourly durations share the same spatial pattern, but with overall enhancement of temperature scaling
(median values of 8.7%/°C at 15-min and 8.2%/°C at 30-min). Given the form of the GEVSS model, however, note that results

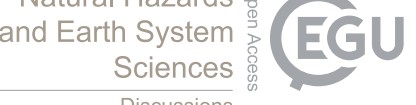



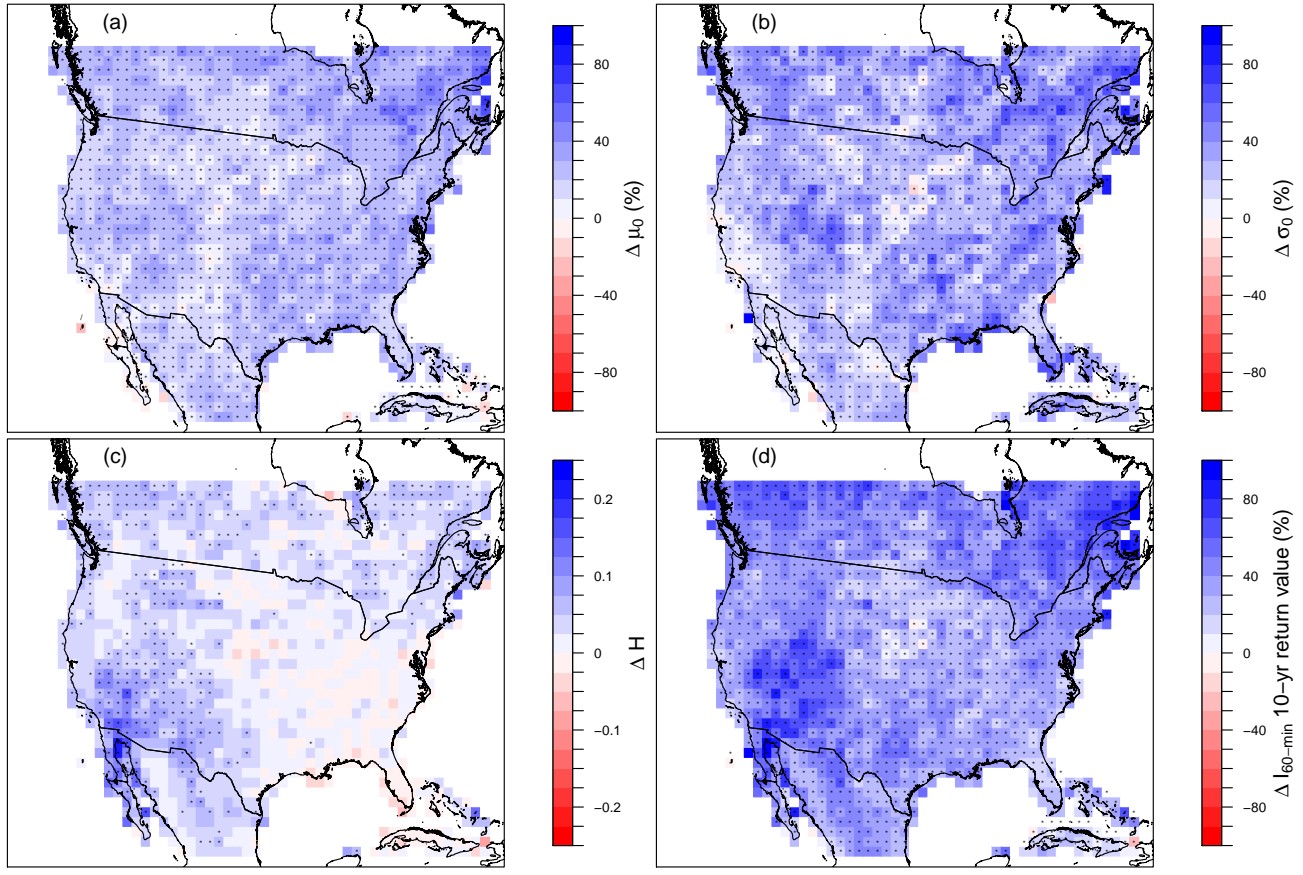

**Figure 9.** Projected changes in GEVSS (a) location $\mu_0$, (b) scale $\sigma_0$, and (c) scaling exponent $H$ parameters, as well as changes in (d) 10-yr return level 1-hr duration rainfall extremes $I_{60-\min,10-\mathrm{yr}}$. For ease of visualization, results are aggregated to a 100-km $\times$ 100-km grid. Values shown are aggregated grid box means; boxes where more than half of the 4-km WRF grids show significant increases (decreases) are marked with a $*$ ($/$).

at sub-daily durations are strongly influenced by projected changes in the scaling exponent parameter $H$ (cf. Figure 9c). At longer durations, median scaling rates are lower (5.6%/°C at 12-hr and 5.0%/°C at 24-hr) and are more spatially uniform, without as strong a gradient between coastal and interior regions. For reference, temperature scaling of GEVSS parameters, rather than return levels, are provided in Figure S6.

5    Given a constant shape $\xi$ parameter between the CTRL and PGW simulations, relative changes in return levels from the GEVSS distribution can vary across return periods only if the dispersion $\sigma_0/\mu_0$ is projected to change. Over North America, Kharin et al. (2018) found that CMIP5 models with parameterized convection project greater intensification of more rare, more intense extreme daily precipitation events than less rare, less intense events. Results from the convection-permitting WRF simulations are shown in Figure 11. Projected changes in $\sigma_0/\mu_0$ are predominantly positive; spatially coherent regions with

10   significant increases are found over 50% of the domain. This leads to a modest dependence of relative changes in return levels





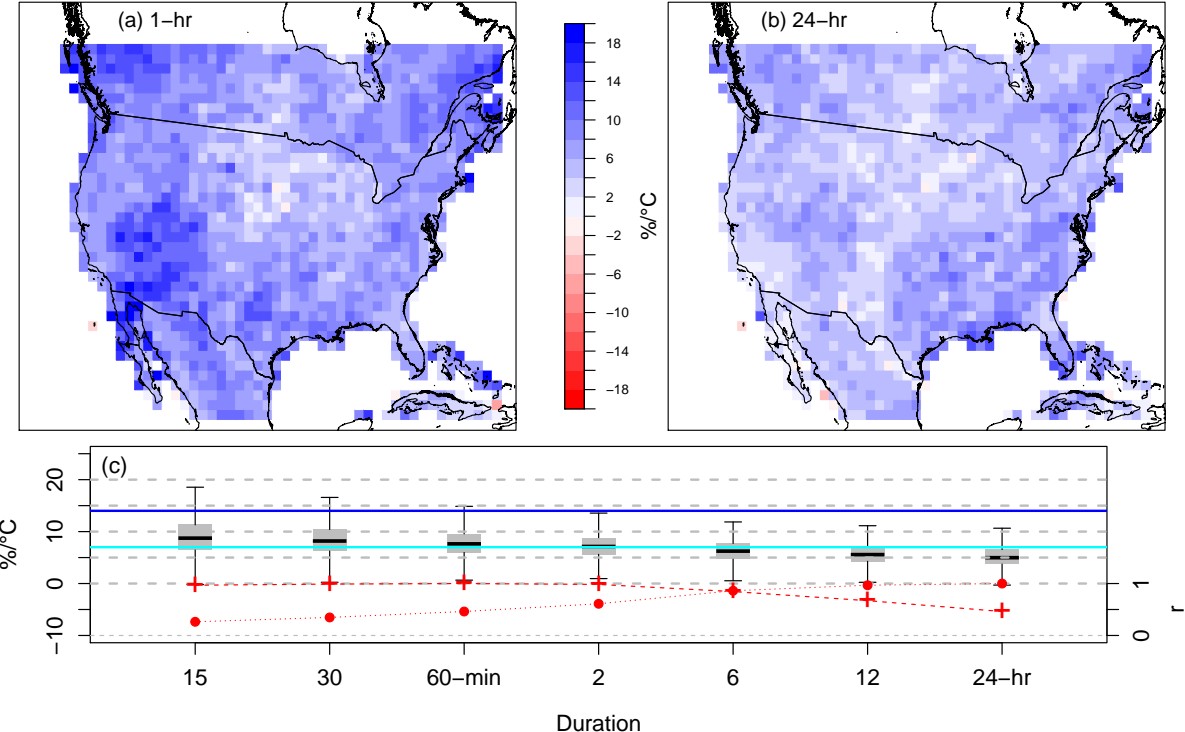

**Figure 10.** Temperature scaling of 10-yr return levels of (a) 1-hr and (b) 24-hr extreme rainfall; values shown are aggregated 100-km × 100-km grid box means. (c) Boxplots summarizing the distribution of temperature scaling values for 10-yr return levels of 15-min to 24-hr duration extreme rainfall. The horizontal line in the middle of each box indicates the median, boxes extend from the lower quartile to the upper quartile, and the whiskers to 1.5 times the IQR. The horizontal cyan (blue) line indicates temperature scaling of 7%/°C (14%/°C.) The red dashed (dotted) lines and the secondary vertical axis indicate the spatial correlation between maps of temperature scaling for each duration and the 1-hr (24-hr) maps.

on return period. When expressed in terms of temperature scaling, median values for the 24-hr duration – the reference duration in the GEVSS model – increase from 4.6%/°C for the 2-yr return period to 5.3%/°C for the 100-yr return period.

# 7   Discussion and conclusions

Annual maximum sub-daily and daily rainfall outputs from continental-scale, decadal simulations of WRF are combined with
5   a parsimonious GEVSS model, which "borrows strength" from multiple durations to improve parameter estimation, to assess changes in IDF curve characteristics over North America. For durations ≥15-min, the GEVSS model leads to IDF curve estimates that are consistent with official curves at TBRG stations in Canada, mirroring findings of Innocenti et al. (2017). In agreement with previous model evaluations that have focused on the US (e.g., Prein et al., 2017c), WRF is found to provide

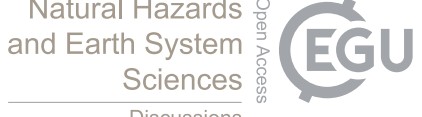



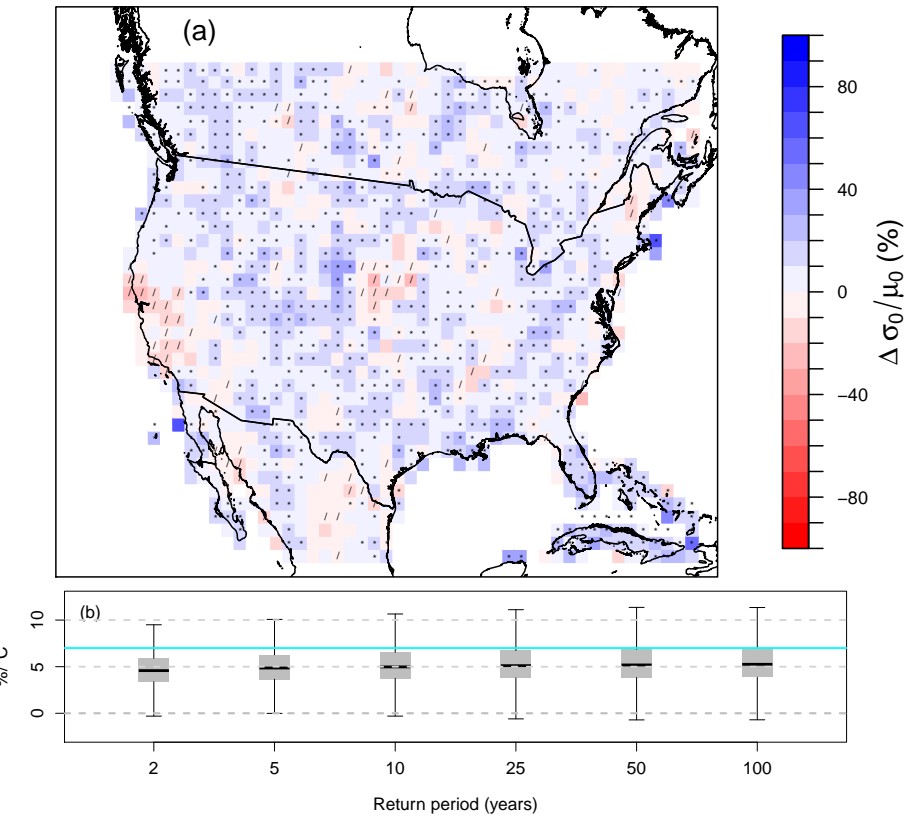

**Figure 11.** Projected changes in (a) GEVSS dispersion $\sigma_0/\mu_0$. Values shown are aggregated 100-km × 100-km grid box means; boxes where more than half of the 4-km WRF grids show significant increases (decreases) are marked with a ∗ (/). (b) Boxplots summarizing the distribution of temperature scaling values for 2-yr to 100-yr return levels of 24-hr duration extreme rainfall; the horizontal cyan line indicates temperature scaling of 7%/°C.

credible simulations of historical annual maximum short-duration rainfall and associated IDF curves over the Canadian portion of the domain.

Projected changes in GEVSS parameters are evaluated using a Bayesian FDR framework. Large portions of the domain experience significant increases in GEVSS location $\mu_0$ ($> 99\%$ of grid points), scale $\sigma_0$ ($> 88\%$), and scaling exponent $H$
5 ($> 39\%$) parameters in the PGW simulation (+3.5°C global warming signal). Increases in each GEVSS parameter are linked to changes in specific characteristics of IDF curves. In particular, IDF curves tend to shift upward (increases in $\mu_0$ and $\sigma_0$), and, with the exception of the eastern US, steepen (increases in $H$). This leads to the largest increases in return levels for short duration extremes. In addition, spatially coherent but small increases in dispersion of the GEVSS distribution are found over more than half of the domain, providing some evidence for return period dependence of future changes in extreme rainfall.
10 When changes in extremes are scaled according to projected regional temperature change, median scaling rates follow the CC



relation at 1-hr to 6-hr durations, which is consistent with CC scaling found by Prein et al. (2017c) for hourly extremes. Spatial patterns and dependence of the temperature scaling relationships on duration are also consistent with results found for station records in Canada (Panthou et al., 2014), although care must be taken due to differences in scaling definitions (e.g., Zhang et al., 2017). Modest sub-CC scaling at longer durations and modest super-CC scaling at sub-hourly durations are found for

other time scales. In general, however, results are consistent with the statement by Pendergrass (2018) that "[s]caling changes of extreme precipitation to the rate of atmospheric moisture increase remains the default null hypothesis", with, as they point out, additional complexity that modifies this default hypothesis. Given that more confidence can be placed on temperature projections, guidance on future changes in IDF curves that is based on regional temperature scaling relationships may provide a simple, actionable path forward for the water resource management and engineering community.

From a statistical perspective, the projected increase in the GEVSS scaling exponent calls into question stationarity assumptions that form the basis for IDF curve projections that rely exclusively on daily model outputs, for example those that statistically disaggregate from the daily to sub-daily time scale using historical temporal scaling relationships (Srivastav et al., 2014). While it is possible that statistical disaggregation methods may have utility for IDF curve projections, those that explicitly condition temporal scaling relationships on atmospheric covariates may be required (Westra et al., 2013b). However, given

short historical records and the small forced change relative to natural variability, developing robust statistical relationships may be difficult.

The parsimony of the GEVSS model used in this study comes at a potential cost; it imposes strong constraints on the form of the IDF curve relationships – and their changes – that can be represented. Checks on GEVSS model assumptions and goodness-of-fit indicate acceptable performance. However, it is possible that less restrictive forms of statistical model

might reveal different future changes. For example, regional quantile regression models for IDF curves (Ouali and Cannon, 2018), including those that explicitly incorporate temporal scaling relationships (Cannon, 2018), are free from the parametric assumptions of the GEVSS model and may provide additional insight into dependence of changes on event rarity and duration. On the other hand, a more flexible model requires estimation of more parameters that may not be as easily interpreted. Trade-offs between model expressiveness and estimation uncertainty are left for future study.

Given differences in model structure, boundary conditions, and domains, it difficult to directly compare results from the continental-scale convection permitting simulations investigated here with those from other midlatitude locations or smaller North American sub-regions. Results are, however, at least qualitatively similar to other studies that have explicitly considered future projections of IDF curves in convection-permitting models. For example, Tabari et al. (2016) found both duration and return period dependence of changes in summer season IDF curves in the CCLM model over central Belgium; similar to

the results from this study, larger intensification was found for shorter durations and longer return periods. Evans and Argueso (2015) found greater increases in WRF simulations of extreme rainfall over the greater Sydney region for longer return periods, but did not identify a clear dependence on duration. Finally, warm season IDF curves based on MM5 simulations over central Alberta by Kuo et al. (2012) were generally projected to shift upward, although dependence on return period and duration was found to be complicated and subject to large uncertainty.



The findings of this study are based on simulations from a single convection-permitting model run over a continental-scale North American domain. While the WRF runs are relatively long for a large, high-resolution domain, the ability to robustly estimate parameters of the GEV distribution, and hence return levels, from 13-yr records is somewhat limited. However, the large warming signal, PGW experimental design, and pooling of durations by the GEVSS model allows changes in IDF curve

characteristics – those driven primarily by thermal effects – to be detected, but raises questions about whether results would be the same under lower levels of warming. Larger samples are needed to reduce sampling uncertainty. In this regard, efforts to run larger multi-model and initial condition ensembles of high-resolution simulations (Gutowski Jr. et al., 2016), including those by convection-permitting models in multiple regions of the world, will lead to more robust characterization of short-duration extremes, sampling of structural uncertainty (e.g., due to differences in microphysics schemes; Singh and O'Gorman,

2014), and regional intercomparisons. Furthermore, including simulations with boundary conditions supplied by global climate models, rather than PGW perturbations of historical data, will allow changes in large-scale dynamics and sensitivity to scenario (e.g., rate of transient warming, role of aerosol emissions as in Lin et al., 2016, etc.) to be assessed. The availability of sub-hourly simulations could also help in investigating the influence of other physical processes on very intense extremes.

Finally, this study did not attempt to link changes in IDF curve characteristics to physical processes. Results using the same

set of WRF simulations by Prein et al. (2017b) indicate large increases in the frequency of mesoscale convective systems (MCSs) and attendant increases in MCS size, total rainfall, and extreme rainfall over much of the domain. A notable exception is central US, a region coincident with the area of lower temperature scaling rates and non-significant changes in $H$ found in this study, which experienced modest decreases in overall MCS frequency, especially for events with low maximum rainfall rates, despite increases in more rare, high rainfall intensity events. Further research investigating changes in storm characteristics

associated with the annual maximum rainfall events that contribute to IDF curve estimates is warranted.

*Data availability.* High-resolution HRCONUS WRF simulations are available for download from Rasmussen and Liu (2017).

*Author contributions.* The first author set the study goals, designed the study, and wrote the paper. Both authors performed data analysis and contributed to the editing process.

*Competing interests.* The authors declare that they have no conflict of interest.

*Acknowledgements.* Kyoko Ikeda kindly facilitated transfer of HRCONUS WRF output from the Research Applications Laboratory, National Center for Atmospheric Research. Comments by Jean-Sébastian Landry on an earlier draft of the manuscript are gratefully appreciated.




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
