# Peer review of "Projected intensification of sub-daily and daily rainfall extremes in convection-permitting climate model simulations over North America: Implications for future Intensity-Duration-Frequency curves"

_Natural Hazards and Earth System Sciences, 2018_

## Referee Comment (RC1) · Anonymous Referee #1 · 13 Nov 2018

Review comments on *nhess-2018-290-manuscript-version1*

**General comments**

Rainfall intensity, duration and frequency (IDF) curves are one of the commonly used tools in engineering and infrastructure design. More intense rainfall events with shorter durations are of more interest because of the more sever damages usually associated with them. However, coarse resolution global climate models (GCMs) are incapable to simulate these rare (with longer return period) events which usually occurs at small/local scales. High resolution regional climate models (RCMs) has been helpful on this regards, higher resolution convection-permitting climate modelling is the trend to better account for the impacts from local geophysical features on intense short duration storms. This paper tried to understand some of the potential impacts on projected IDF curves from convection-permitting climate simulations over the North America. Conclusions from this study would contribute to improvement of future projected IDF curves, although a lot of further research is still needed, as the authors pointed out.

The paper was well-written with logic reasoning in the text, limitations/disclaimers clearly stated, and potential further research identified in the conclusions. Understanding the data availability issues, I would really like to see projected IDF curves based on convection-permitting model results driven by the GCM results under the IPCC RCPs (opposed to the PGW scenario in this study) using this GEVSS method in future studies.

**I would suggest this paper be published with the following minor technical corrections.**

*1 minor typo correction*:

Line 25 on page 23: add "is" between "it" and "difficult".

*1 question*:

The authors mentioned the difference between their GEVSS and conventional (e.g. ECCC) methodology for IDF curves, are the IDF curves from this study ready for engineers to use for their infrastructure designing and planning? Do the authors see any challenges to persuade engineers to use their new IDF curves from this study, instead of those based on conventional method, which have been used for ages, if not longer?

---

## Author Comment (AC1) · 20 Nov 2018

Below we present replies (Reply) to the reviewer comments (Comment) below.

—-

Comment: 1 minor typo correction: Line 25 on page 23: add "is" between "it" and "difficult".

[Figure]

Reply: This typo will be fixed in the revised manuscript.

—-

Comment: 1 question: The authors mentioned the difference between their GEVSS and conventional (e.g. ECCC) methodology for IDF curves, are the IDF curves from this study ready for engineers to use for their infrastructure designing and planning? Do the authors see any challenges to persuade engineers to use their new IDF curves from this study, instead of those based on conventional method, which have been used for ages, if not longer?

Reply: The GEVSS model allows for nonzero shape parameter, but constrains all quantile curves to share the same slope. This reduces the number of parameters relative to the official ECCC IDF curves, while keeping the main characteristics of the conventional approach, but also allowing some flexibility in terms of the ability to model heavy tailed distributions. Figures 7 and 8 compare model performance statistics between GEVSS estimates of IDF curves relative to the official ECCC IDF v2.30 values. Systematic and unsystematic errors are, in general, very low, which suggests that the two methodologies provide similar quantile estimates. While it beyond the scope of the paper to make recommendations about the use of alternative methods operationally, the revised manuscript will include some additional discussion on potential advantages/disadvantages of the GEVSS model relative to the conventional ECCC approach.

---

## Referee Comment (RC2) · Anonymous Referee #2 · 23 Nov 2018

Dear All:

Overall, I found the manuscript to be reasonably well polished with minor editorial and presentation issues (see below). If there are any thing "major" to comment on: The paper, at times, strikes me to be on the technical side which may irk broader audiences (i.e. stakeholder, data users, climate modellers with less background in extreme value theory). That said, I appreciate the paper's conciseness. While detail discussion is

technical, the conclusions is very accessible to the general audience. Overall, I think the paper is acceptable with some minor presentation changes

Specific comment and questions:

Abstract lines 13-15: What are the physical meanings of changes to GEVSS parameters? That information may be too technical for an abstract.

Table 1: A horizontal separator for each dataset item may make the Table somewhat easier to follow.

Pg 12 Lines 8: "from the 488 IDF curve TBRG stations shown in Figure 1." ← This may require rephrasing. Perhaps something like "... from the 488 IDF curves derived from TBRG stations; the station locations are shown in Figure 1".

Figure 7 captions/Pg 16 line 15: "Results are compared with the IDF curves disseminated by ECCC" should be mentioned in the caption as well for the sake of clarity.

Page 16 Lines 30-31 to Page 17 lines ∼10: This is an example why overly technical discussion may bury important end user result. It will help the general reader if "For example, if the goal is to assess whether there is evidence for a steepening of IDF curve in the future" to stand out from the rest of the discussion. Perhaps a paragraph break will help here?

Best Regards Anonymous Reviewer

---

## Author Comment (AC2) · 6 Dec 2018

Below we present replies (Reply) to the reviewer comments (Comment) below.

—-

Comment: Overall, I found the manuscript to be reasonably well polished with minor editorial and presentation issues (see below). If there are any thing "major" to comment

on: The paper, at times, strikes me to be on the technical side which may irk broader audiences (i.e. stakeholder, data users, climate modellers with less background in extreme value theory). That said, I appreciate the paper's conciseness. While detail discussion is technical, the conclusions is very accessible to the general audience. Overall, I think the paper is acceptable with some minor presentation changes

Reply: The reviewer's point about striking a balance between technical detail and accessibility, while still being concise, is well taken. The original goal was to try to write a paper that could be read and found useful by multiple audiences – end users, statistical climatologists, climate modelers/climate scientists – without compromising too much in any particular aspect. The edits and suggestions made below should help with accessibility and clarity. Where possible, edits to the final draft will keep this overall comment in mind as well.

—-

Comment: Abstract lines 13-15: What are the physical meanings of changes to GEVSS parameters? That information may be too technical for an abstract.

Reply: The abstract in the revised manuscript will be modified to better link information on projected changes in the GEVSS parameters to the physical changes in rainfall intensity that each implies.

—-

Comment: Table 1: A horizontal separator for each dataset item may make the Table somewhat easier to follow.

Reply: Horizontal separators will be added to the revised manuscript.

—-

Comment: Pg 12 Lines 8: "from the 488 IDF curve TBRG stations shown in Figure 1."
<- This may require rephrasing. Perhaps something like "... from the 488 IDF curves

derived from TBRG stations; the station locations are shown in Figure 1".

Reply: Agreed. The suggested text will be added to the revised manuscript.

—-

Comment: Figure 7 captions/Pg 16 line 15: "Results are compared with the IDF curves disseminated by ECCC" should be mentioned in the caption as well for the sake of clarity.

Reply: Agreed. This will be added to the Figure caption.

—-

Comment: Page 16 Lines 30-31 to Page 17 lines ∼10: This is an example why overly technical discussion may bury important end user result. It will help the general reader if "For example, if the goal is to assess whether there is evidence for a steepening of IDF curve in the future" to stand out from the rest of the discussion. Perhaps a paragraph break will help here?

Reply: A paragraph break here would help and will be added to the revised manuscript.

---

## Referee Comment (RC3) · Anonymous Referee #3 · 26 Dec 2018

Intensity, duration and frequency curves of extreme precipitation are important tools for designing water management structures and other engineering design applications. However, unavailability of long data both in the observation and high resolution climate model simulations is a major issue in estimating precipitation quantiles for longer return periods. The paper attempts to address this "key problem of shortage of data" by pooling the data from different durations to fit a robust GEV distribution. The GEVSS model based upon the simple scaling hypothesis seems to provide a nice framework

for assessing changes in specific characteristics of IDF curves .

The paper is nicely written with the strengths and weaknesses of the method discussed in the text. However, anticipating a wide reach of the paper, explanation to a couple of technical words used in the paper is requierd.

I suggest that this paper be published with the following minor comments:

1) How does the station-specific estimates based upon GEVSS compare with the estimates based upon the regional frequency analysis?

2) As mentioned in the line 12 of the manuscript, what interpolation method did you use to interpolate the data from grid points to the station locations? Did you try considering different interpolation schemes to see which scheme introduces the least error?

3) Are there any significance tests for the statistics MLAR and MALAR? If yes, it would be better to include results of the tests in figures 5,7-8.

4) Please explain in detail how did you perform "permutation test" mentioned in line 26.

5) Please explain the term "Pseudo-global warning".

6) please explain the term "convection-permitting".

---

## Author Comment (AC3) · 7 Jan 2019

Below we present replies (Reply) to the reviewer comments (Comment) below.

—-

Comment 1): How does the station-specific estimates based upon GEVSS compare with the estimates based upon the regional frequency analysis?

Reply: The reviewer raised an interesting question regarding the performances of the GEVSS model compared to other classical IDF estimation methods, such as Regional Frequency Analysis (RFA). The direct comparison of station-specific GEVSS IDF with RFA IDF would in fact allow to evaluate more thoroughly the GEVSS performances and refine the validation of IDF simple scaling models that have been undertaken in some previous studies [e.g., Blanchet et al. 2016, Innocenti et al. 2017, Boukhelifa et al. 2018].

However, a more sophisticated methodology should be considered to accurately define a meaningful comparison between GEVSS and RFA IDF. For instance, the estimation of RFA IDF relies on the identification of homogeneous geographical regions that allow to pool stations with similar characteristics. This can be achieved by various methods [e.g., Grimaldi et al, 2011; Hosking and Wallis, 1997; among others] which yield different performances in different specific analyses. While it is beyond the scope of this study to perform such additional analyses, the revised manuscript will stress the importance of considering alternative methods for the estimation and validation of IDF curves in practical situations.

Boukhelifa M, Meddi M, Gaume E. Integrated Bayesian Estimation of Intensity-Duration-Frequency Curves: Consolidation and Extensive Testing of a Method. Water Resources Research. 2018 Oct; 54(10):7459-77

Grimaldi S, Kao S-C, Castellarin A, Papalexiou S M, Viglione A, Laio F, Aksoy H and Gedikli A(2011) Statistical Hydrology. Peter Wilderer (ed.) Treatise on Water Science, vol. 2, pp. 479–517 Oxford: Academic Press.

Hosking, J.R.M., Wallis, J.R., 1997. Regional Frequency Analysis: An Approach Based on L-moments. Cambridge, UK, 244pp

—-

Comment 2) As mentioned in the line 12 of the manuscript, what interpolation method

did you use to interpolate the data from grid points to the station locations? Did you try considering different interpolation schemes to see which scheme introduces the least error?

Reply: As suggested by the reviewer, the method used to associate points and grid box estimates at different spatial resolutions inevitably affects the results of gridded dataset evaluation. Moreover, it is generally difficult to quantitatively assess the effects of this resolution mismatch and various interpolation methods should be compared for different errors statistics, including those considered in our study. However, a comprehensive evaluation of the relative contribution of the resolution mismatch to the overall difference between grid and point rainfall estimate is beyond the scope of the paper. For sake of simplicity, stations were thus associated to the corresponding (overlapping) grid box of each gridded dataset without the use of additional interpolation methods. To clarify this point the following sentence will be added to the text (p. 12 l.17): "Locations were identified by station coordinates and stations were associated to the overlapping (nearest centroid) grid box within each dataset."

—-

Comment 3) Are there any significance tests for the statistics MLAR and MALAR? If yes, it would be better to include results of the tests in figures 5,7-8.

Reply: No significance test was initially considered for the analysis of MLAR and MALAR. Mention of the field significance [Wilks, 2006] of permutation test results could be added to the revised manuscript to stress the significance of the two aggregated error statistics.

Wilks, D.S., On "field significance" and the false discovery rate. Journal of Applied Meteorology and Climatology. 2006 Sep; 45(9):1181-9.

—-

Comment 4) Please explain in detail how did you perform "permutation test" mentioned

in line 26.

Reply: The description of the applied permutation test (p. 12 l.26) will be modified to: "A permutation test was used to estimate the statistical significance of the RE values. Assuming the equality of station and grid box distribution of X, the permutation test considers a null hypothesis of quantile equality. Under this null hypothesis, station and grid box annual maxima were pooled for each given location. Pooled observations were then randomly reassigned to two permutation resamples (i.e. two annual maxima series of shuffled observations) having equal length than the station and grid box original (unpermuted) samples. For each station-grid box pair, the distribution of the RE statistics were thus approximated on 5000 random permutation resamples and the p-value was computed as the fraction of resamples generating RE absolute values equal or larger than those observed on the original annual maxima samples."

—-

Comment 5) Please explain the term "Pseudo-global warning".

Reply: The description of the pseudo-global-warming technique [e.g., Rasmussen et al. 2017 and references therein] will be added to the revised manuscript.

Rasmussen, K. L., Prein, A. F., Rasmussen, R. M., Ikeda, K., and Liu, C.: Changes in the convective population and thermodynamic environments in convection-permitting regional climate simulations over the United States, Climate Dynamics, https://doi.org/10.1007/s00382-017-4000-7, 2017.

—-

Comment 6) please explain the term "convection-permitting". Reply: A brief explanation of the term will be added in Sect. 2 of the revised manuscript, as well as the reference to key discussions about convection-permitting modeling techniques [e.g., Prein et al. 2015 and Kendon et al., 2017].

Kendon EJ, Ban N, Roberts NM, Fowler HJ, Roberts MJ, Chan SC, Evans JP, Fosser

G, Wilkinson JM. Do convection-permitting regional climate models improve projections of future precipitation change?. Bulletin of the American Meteorological Society. 2017 Jan; 98(1):79-93.

Prein AF, Langhans W, Fosser G, Ferrone A, Ban N, Goergen K, Keller M, Tölle M, Gutjahr O, Feser F, Brisson E. A review on regional convection-permitting climate modeling: Demonstrations, prospects, and challenges. Reviews of Geophysics. 2015 Jun; 53(2):323-61.
* * *

---

## Referee Report (RR1)

General comments:

Thank you for making the changes. Here are some newer details that I have spotted. The minor comment is very minor, but the major comment may require some minor checking.

Major Comment:

Pg 12, lines 19-22: You could probably do some quick areal reduction factor (ARF) calculations to show how good is to assume high resolution gridded data can be compared with point data without correction. I do not know the ARF guidelines for US or Canada, but UK ARF guidelines (Kjeldsen 2007; see Chapter 4, free PDF is available at the referenced website) suggest with WRF, CMORPH and NSWEP grid at 39N (Washington DC) to have ARF ~ 0.9, 0.85, 0.8 respectively for 1hr precipitation. For 24h precipitation, ARF > 0.95. Of course, ARF themselves are approximations and region specific, but it is probably a good idea to comment on that how may affect your results. For example for your Fig 5, failure to account for areal averaging may push your MLAR curves downward; WRF's MLAR for 1h is negative about -0.08 (?); assuming UK ARF, log10(0.9) ~ -0.05, so your WRF results may actually look better than the plot indicates. Overall, your comment (Pg. 14, lines 23-24) that WRF is probably better than CMORPH in the representation of the current-climate rainfall extremes remains true (in fact WRF may be better than you think).

Minor Comment:

Equation 6: Define the meaning of U (uniform distribution) and N (normal distribution) and their bracketed parameters; some readers may not be familiar with that notation style.

Kjeldsen, T.R.. (2007). *The revitalised FSR/FEH rainfall-runoff method*. Centre for Ecology & Hydrology. Wallingford. Retrieved from https://www.ceh.ac.uk/services/flood-estimation-handbook

---

## Referee Report (RR2)

The authors have definitely improved the manuscript and have addressed most of my comments in the revised manuscript. The revised manuscript can be accepted for publication subject to two minor suggestions.

1.  The authors had replied to my previous comment #3 in the discussion. But I do not see any mentioning of it in the manuscript. I apologize if I missed it in the manuscript. I request authors to make it clear if they have included the response to the revised manuscript. **It should be clearly seen in the text if a significance test for MLAR and MALAR have been added. If no such significance test has been added, the authors should make it clear as to why a significance test cannot added?**

2.  The authors have added the text "Grid points in each dataset were matched with the nearest neighboring station." in line 25 of page 12.  As far as I understand, the grid point nearest to the station is picked for computing "RE", "MLAR" and "MALAR". However, text in the last paragraph on page 13 mentions "station and grid box annual maxima were pooled for each given location". This confuses me. Did you include "all" grid points surrounding the station to pool with the station data or just "one" grid point nearest to the station? Please make it clear in the text.

---

## Author Response (AR2)

The authors thank the referees for their useful and constructive comments. Replies are provided below in red.
* * *
Referee #1:

General comments:

Thank you for making the changes. Here are some newer details that I have spotted. The minor comment is very minor, but the major comment may require some minor checking.

Major Comment:

Pg 12, lines 19-22: You could probably do some quick areal reduction factor (ARF) calculations to show how good is to assume high resolution gridded data can be compared with point data without correction. I do not know the ARF guidelines for US or Canada, but UK ARF guidelines (Kjeldsen 2007; see Chapter 4, free PDF is available at the referenced website) suggest with WRF, CMORPH and NSWEP grid at 39N (Washington DC) to have ARF ~ 0.9, 0.85, 0.8 respectively for 1hr precipitation. For 24h precipitation, ARF > 0.95. Of course, ARF themselves are approximations and region specific, but it is probably a good idea to comment on that how may affect your results. For example for your Fig 5, failure to account for areal averaging may push your MLAR curves downward; WRF's MLAR for 1h is negative about -0.08 (?); assuming UK ARF, log10(0.9) ~ -0.05, so your WRF results may actually look better than the plot indicates. Overall, your comment (Pg. 14, lines 23-24) that WRF is probably better than CMORPH in the representation of the current-climate rainfall extremes remains true (in fact WRF may be better than you think).

An analysis of area-to-point corrected MLAR and MALAR statistics has been added to the text and supplemental material. Specifically, Figures S3-S5 provide analogues of Figures 5, 7, and 8 (MLAR and MALAR plots) but with prior adjustment of the gridded data quantiles using inverse areal reduction factors based on Kjeldsen (2007). As the referee surmised, the WRF results now outperform CMORPH and are essentially unbiased for durations from 15-min to 6-hr.

Minor Comment:

Equation 6: Define the meaning of U (uniform distribution) and N (normal distribution) and their bracketed parameters; some readers may not be familiar with that notation style.

Done.

Kjeldsen, T.R.. (2007). The revitalised FSR/FEH rainfall-runoff method. Centre for Ecology & Hydrology. Wallingford. Retrieved from https://www.ceh.ac.uk/services/flood-estimation-handbook

The authors have definitely improved the manuscript and have addressed most of my comments in the revised manuscript. The revised manuscript can be accepted for publication subject to two minor suggestions.

1. The authors had replied to my previous comment #3 in the discussion. But I do not see any mentioning of it in the manuscript. I apologize if I missed it in the manuscript. I request authors to make it clear if they have included the response to the revised manuscript. It should be clearly seen in the text if a significance test for MLAR and MALAR have been added. If no such significance test has been added, the authors should make it clear as to why a significance test cannot added?

Confidence intervals have been added to the MLAR/MALAR plot for the empirical quantiles (Figure 5), with the following explanation added to the text:

Values are accompanied by 95% confidence intervals estimated based on 1000 bootstrap samples drawn from the series of annual maxima at each location.

Similarly, 95% credible intervals have been added to the MLAR and MALAR plots for the GEVSS estimates (Figures 7 and 8), with the following explanation added to the text:

In all cases, posterior means and 95% credible intervals for the MLAR and MALAR statistics are estimated from the posterior distributions of the GEVSS parameters and the resulting return levels.

Results are used to infer statistical significance.

2. The authors have added the text "Grid points in each dataset were matched with the nearest neighboring station." in line 25 of page 12. As far as I understand, the grid point nearest to the station is picked for computing "RE", "MLAR" and "MALAR". However, text in the last paragraph on page 13 mentions "station and grid box annual maxima were pooled for each given location". This confuses me. Did you include "all" grid points surrounding the station to pool with the station data or just "one" grid point nearest to the station? Please make it clear in the text.

One grid point nearest to the station was used in all cases. The text has been reworded to make it clear that "pooling" in this case means combining durations for a single location, rather than any pooling of different locations in space:

Under this null hypothesis, station and grid box **annual maxima at a given location are combined into a single sample**. The combined data are then randomly reassigned to two permutation resamples (i.e., two series of shuffled annual maxima) having equal length as the original, unpermuted station and grid box samples. For each station-grid box pair, the distribution of the RE statistics is approximated using 5000 random permutation resamples and the p-value is computed as the fraction of resamples generating RE absolute values equal to or larger than those observed on the original annual maxima samples.